# Olfactory channels associated with the *Drosophila* maxillary palp mediate short- and long-range attraction

**Hany KM Dweck[1]\*†, Shimaa AM Ebrahim[1]†, Mohammed A Khallaf[1], Christopher Koenig[1], Abu Farhan[1]§, Regina Stieber[1], Jerrit Weißflog[2], Aleš Svatoš[2], Ewald Grosse-Wilde[1], Markus Knaden[1]\*‡, Bill S Hansson[1]\*‡**

[1]Department of Evolutionary Neuroethology, Max Planck Institute for Chemical Ecology, Jena, Germany; [2]Mass Spectrometry Group, Max Planck Institute for Chemical Ecology, Jena, Germany

**Abstract** The vinegar fly *Drosophila melanogaster* is equipped with two peripheral olfactory organs, antenna and maxillary palp. The antenna is involved in finding food, oviposition sites and mates. However, the functional significance of the maxillary palp remained unknown. Here, we screened the olfactory sensory neurons of the maxillary palp (MP-OSNs) using a large number of natural odor extracts to identify novel ligands for each MP-OSN type. We found that each type is the sole or the primary detector for a specific compound, and detects these compounds with high sensitivity. We next dissected the contribution of MP-OSNs to behaviors evoked by their key ligands and found that MP-OSNs mediate short- and long-range attraction. Furthermore, the organization, detection and olfactory receptor (Or) genes of MP-OSNs are conserved in the agricultural pest *D. suzukii*. The novel short and long-range attractants could potentially be used in integrated pest management (IPM) programs of this pest species.

**\*For correspondence:** hdweck@ice.mpg.de (HKMD); mknaden@ice.mpg.de (MK); hansson@ice.mpg.de (BSH)

†These authors contributed equally to this work
‡These authors also contributed equally to this work

**Present address:** §Department of Cell Biology, Molecular Neurobiology Program, Skirball Institute of Biomolecular Medicine, New York University School of Medicine, New York, United States

## Introduction

Like all insects, the vinegar fly, *D. melanogaster*, is equipped with two peripheral olfactory organs, the antenna and maxillary palp. The antenna, the main olfactory organ, is covered with four types of sensilla: basiconic, trichoid, intermediate and coeloconic. These four sensillum types house olfactory sensory neuron (OSN) types responding to different kinds of chemical stimuli and thus serve distinct chemosensory functions. In contrast, the palp has only three different subtypes of basiconic sensilla, each housing two MP-OSNs. Because of the overlapping response spectra between MP- and antennal OSNs (Ant-OSNs) (*de Bruyne et al., 1999*; *2001*) as well as the location of the maxillary palp in close vicinity to the labellum, the main taste organ in flies, a function connected to taste enhancement has been suggested for the MP-OSNs (*Shiraiwa et al., 2008*). However, taste enhancement would be a very general function for six types of MP-OSNs expressing seven different odorant receptors (Ors). In our previous work we presented data on the importance of Or71a, which is expressed in the maxillary palp sensillum pb1B, in proxy detection of dietary antioxidants (*Dweck et al., 2015*). It is not yet known whether the other MP-OSNs are also dedicated to detect specific ecologically relevant chemical compounds, and if so, what the ecological importance of these compounds is.

In several other insects, MP-OSNs are involved in detection of specific chemical compounds that are not covered within the receptive range of Ant-OSNs. For example, in both the hawk moth *Manduca sexta* and the African malaria mosquito *Anopheles gambiae*, $CO_2$ detection is primarily mediated via maxillary and/or labial palp OSNs (*Thom et al., 2004*; *Lu et al., 2007*). Mammals are also

known to possess several peripheral olfactory organs. In mouse, e.g., the main olfactory epithelium is complemented with the vomeronasal organ, the septal organ and the Grueneberg ganglion, each having distinct functions (reviewed in *Knaden and Hansson, 2014*). The presence of specific functions in different olfactory organs in other insects and in mammals suggests that the maxillary palp may also be involved in the detection of specific chemicals in *Drosophila*.

In the present study, we present a systematic electrophysiological examination of MP-OSNs of *D. melanogaster* using 52 different complex odor sources containing more than 11,300 chemical compounds. We find that each MP-OSN is either the sole or the primary detector of a specific chemical compound and that the maxillary palp contains independent and important olfactory channels that mediate both short- and long-range attraction. Finally, we find that the organization, detection and Or genes of MP-OSNs are conserved in the agricultural pest *D. suzukii*, and identify novel short and long-range attractants that could potentially be used in IPM programs of this pest species.

## Results and discussion

### Screen for novel natural ligands for MP-OSNs

Although extensive work has been done on the olfactory sense of the vinegar fly, *D. melanogaster*, none (e.g. for pb3A-OR59c, Pb2A-Or33c) or very few ligands have been identified for the different MP-OSNs (*de Bruyne et al., 1999*, *Goldman et al., 2005*; *Marshall et al., 2010*). In addition, the previously identified ligands activate Ant-OSNs as well as MP-OSNs and have been shown to be much better ligands for Ant-OSNs (i.e. require high concentrations to activate MP-OSNs) (*de Bruyne et al., 1999*; *Hallem and Carlson, 2006*). This suggests that the best ligands of the different MP-OSNs have not yet been identified. Towards this end, we screened each of the six MP-OSNs with headspace collections from 52 different complex, ecologically relevant odor sources using GC-SSR (*Figure 1A,B* and *Figure 1—figure supplement 1–6*, *Figure 1—source data 1*). These odor sources included 34 fruits, seven microbes, and eleven types of mammal feces. Our GC-SSR measurements revealed that each of the tested headspace collections triggered a response from at least one palp OSN type. Fecal, fruit and microbial volatiles elicited responses from six, five and two palp OSN types, respectively. The pb2B MP-OSN was activated exclusively by fecal volatiles, whereas the other five types were activated by fruit, microbial, and fecal volatiles.

These large-scale GC-SSR experiments allowed us to test whether the 52 headspace collections of fruits, microbes and mammal feces are separated in the neural space of the maxillary palp. We performed a nonmetric multidimensional scaling (NMDS) based on a presence/absence matrix for the SSR active peaks across the tested samples using Bray-Courtis dissimilarity. This analysis indicated that the 52 headspace collections were separated into three distinct groups; one group for fruit samples, another group for microbial samples and the last group for fecal samples (*Figure 1C*, *Figure 1—source data 1*). The significance of the differences between these three groups was assessed by the analysis of similarity (ANOSIM) score (R = 0.61, p<0.0001). These results suggest that the information provided by the different MP-OSNs is sufficient to categorize fruits, microbes, and feces.

The physiologically active peaks from each extract were then identified via GC-mass spectroscopy (GC-MS) and co-injection with synthetic standards, which were purchased except for 5-hexen-3-one and butyl-3-hydroxy butyrate, which were synthesized in house (see Materials and methods). The total number of distinguishable flame ionization detection (FID) peaks in the samples was 11326, of which only 328 FID peaks elicited responses (*Figure 1B*, *Figure 1—figure supplement 7*, and *Figure 1—figure supplement 8*). 225 of these peaks corresponded to 20 different compounds (*Table 1*). The remaining peaks corresponded to eight different compounds, which remain unidentified because their mass spectra did not match that of any reference compound. The identified compounds belonged to four different chemical classes: alcohols, esters, phenols and ketones. Six of the physiologically active compounds occurred in most extracts, whereas the other 22 compounds were extract specific. Phenol and 4-methylphenol occurred exclusively in fecal extracts (*Figure 1B*, *Figure 1—figure supplement 4*).

We next compared the distribution of the ligands recognized by different MP-OSNs in an odor space of 32 DRAGON descriptors (i.e. physicochemical properties such as number of benzene-like rings and number of double bonds), which were previously selected by *Haddad et al. (2008)*. The

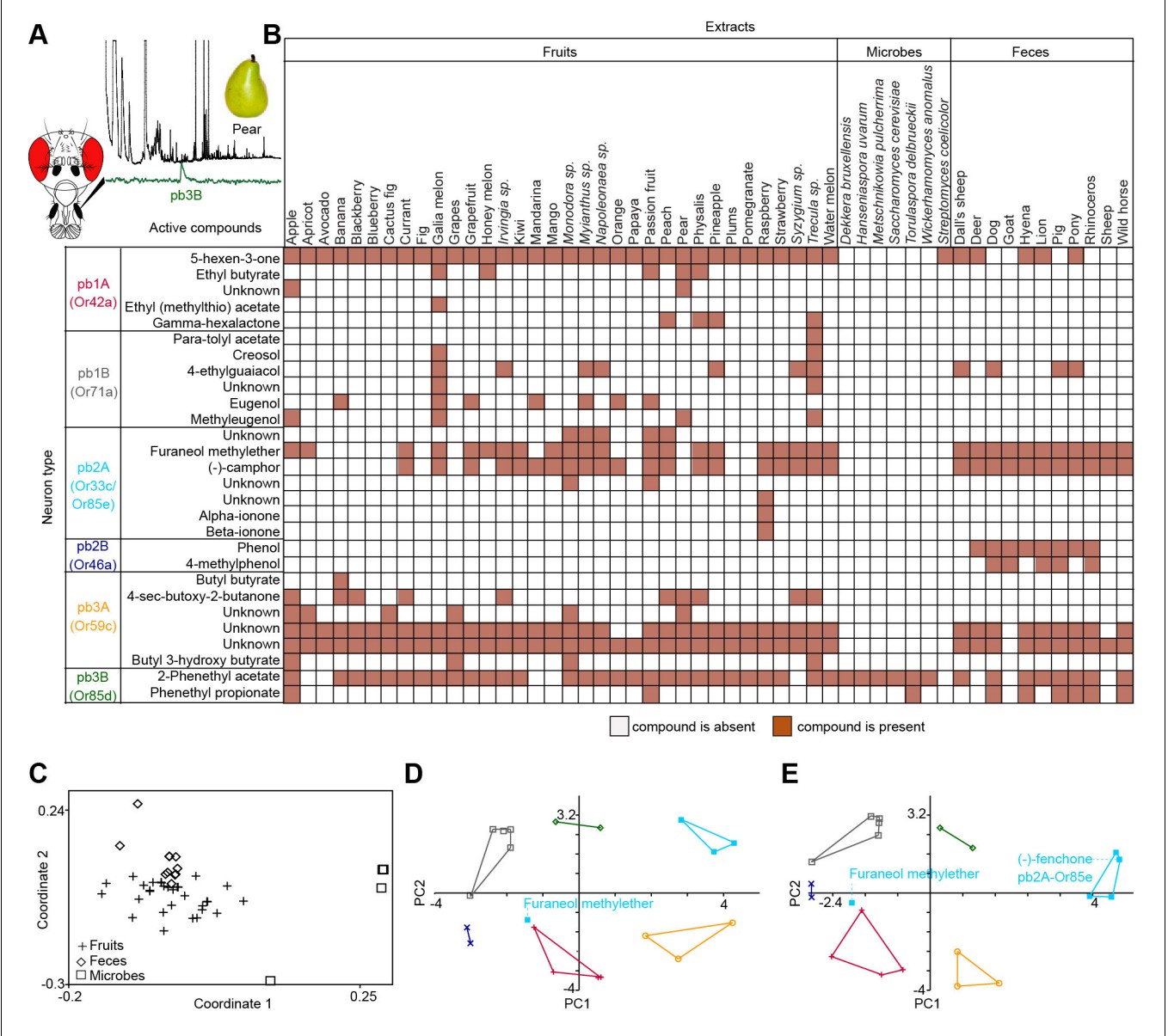

**Figure 1.** Screen for novel natural ligands for MP-OSNs. (A) Representative gas chromatography-linked single sensillum measurement (GC-SSR) from pb3B (green trace) stimulated with headspace extract of pear (black trace). (B) Presence/Absence matrix of the physiologically active compounds identified from the different headspace extracts for each MP-OSN in the GC-SSR experiments (i.e. each filled box represents not only the presence of this odor in a specific fruit, but also a physiological response in GC-SSR recordings). (C) NMDS plot based on a presence/absence matrix for the active peaks across the tested samples. (D) PCA plot showing the distribution of the ligands recognized by MP-OSNs in a 32-dimensional odor space. PC1 and PC2 explain 23% and 22% of the variance, respectively. (E) PCA plot showing the distribution of the ligands recognized by MP-OSNs and (-)-fenchone (the main ligand of Or85e-expressing OSNs) in a 32-dimensional odor space. PC1 and PC2 explain 24% and 21% of the variance, respectively.

The following source data and figure supplements are available for figure 1:

**Source data 1.** Presence and absence data as well as physicochemical properties of all tested odor samples, that were used to calculate the NMDS plot and the PCAs in *Figure 1*.

**Figure supplement 1.** Responses of pb1A OSNs type to physiologically active compounds in different extracts.

**Figure supplement 2.** Responses of pb1B OSNs type to physiologically active compounds in different extracts.

**Figure supplement 3.** Responses of pb2A OSNs type to physiologically active compounds in different extracts.

*Figure 1 continued on next page*

*Figure 1 continued*

**Figure supplement 4.** Responses of pb2B OSNs type to physiologically active compounds in different extracts.

**Figure supplement 5.** Responses of pb3A OSNs type to physiologically active compounds in different extracts.

**Figure supplement 6.** Responses of pb3B OSNs type to physiologically active compounds in different extracts.

**Figure supplement 7.** GC-MS chromatographs showing number of FID peaks in each fruit sample.

**Figure supplement 8.** GC-MS chromatographs showing number of FID peaks in each microbial (**A**) and fecal (**B**) sample.

32 descriptors were then normalized using *z*-scrores and visualized in a two-dimensional principal component analysis (PCA) plot using variance-covariance matrix (*Figure 1D*, *Figure 1—source data 1*). In this odor space, odors with similar descriptors mapped close to each other, whereas odors with diverse descriptors distributed widely. Indeed, compounds that activated different MP-OSNs differed in their descriptors and, hence, distributed widely in the two-dimensional odor space. Compounds, however, that activated the same MP-OSN clustered together except for the ligands recognized by pb2A (*Figure 1D*).

Three out of four identified ligands for pb2A grouped close to each other, whereas the fourth ligand (furaneol methylether) spaced very widely. This could be explained by the fact that pb2A is the only MP-OSN that expresses two olfactory receptors, Or33c and Or85e (*Couto et al., 2005*; *Goldmann et al., 2005*). In order to predict which of these two receptors could detect which of the pb2A ligands, we included (-)-fenchone, a previously identified best ligand for Or85e

**Table 1.** List of physiologically active compounds identified for MP-OSNs including their Chemical Abstract Service numbers (CAS no.).

| Compound | CAS no. |
| --- | --- |
| 5-hexen-3-one | 24253-30-3 |
| Ethyl butyrate | 105-54-4 |
| Ethyl (methylthio) acetate | 4455-13-3 |
| Gamma-hexalactone | 695-06-7 |
| Para-tolyl acetate | 140-39-6 |
| Creosol | 93-51-6 |
| 4-ethylguaiacol | 2785-89-9 |
| Eugenol | 97-53-0 |
| Methyleugenol | 93-15-2 |
| Furaneol methylether | 4077-47-8 |
| (-)-camphor | 464-48-2 |
| Alpha-ionone | 127-41-3 |
| Beta-ionone | 14901-07-6 |
| Phenol | 108-95-2 |
| 4-methylphenol | 106-44-5 |
| Butyl butyrate | 109-21-7 |
| 4-sec-butoxy-2-butanone | 106-44-5 |
| Butyl 3-hydroxy butyrate | 53605-94-0 |
| 2-Phenethyl acetate | 103-45-7 |
| Phenethyl propionate | 122-70-3 |

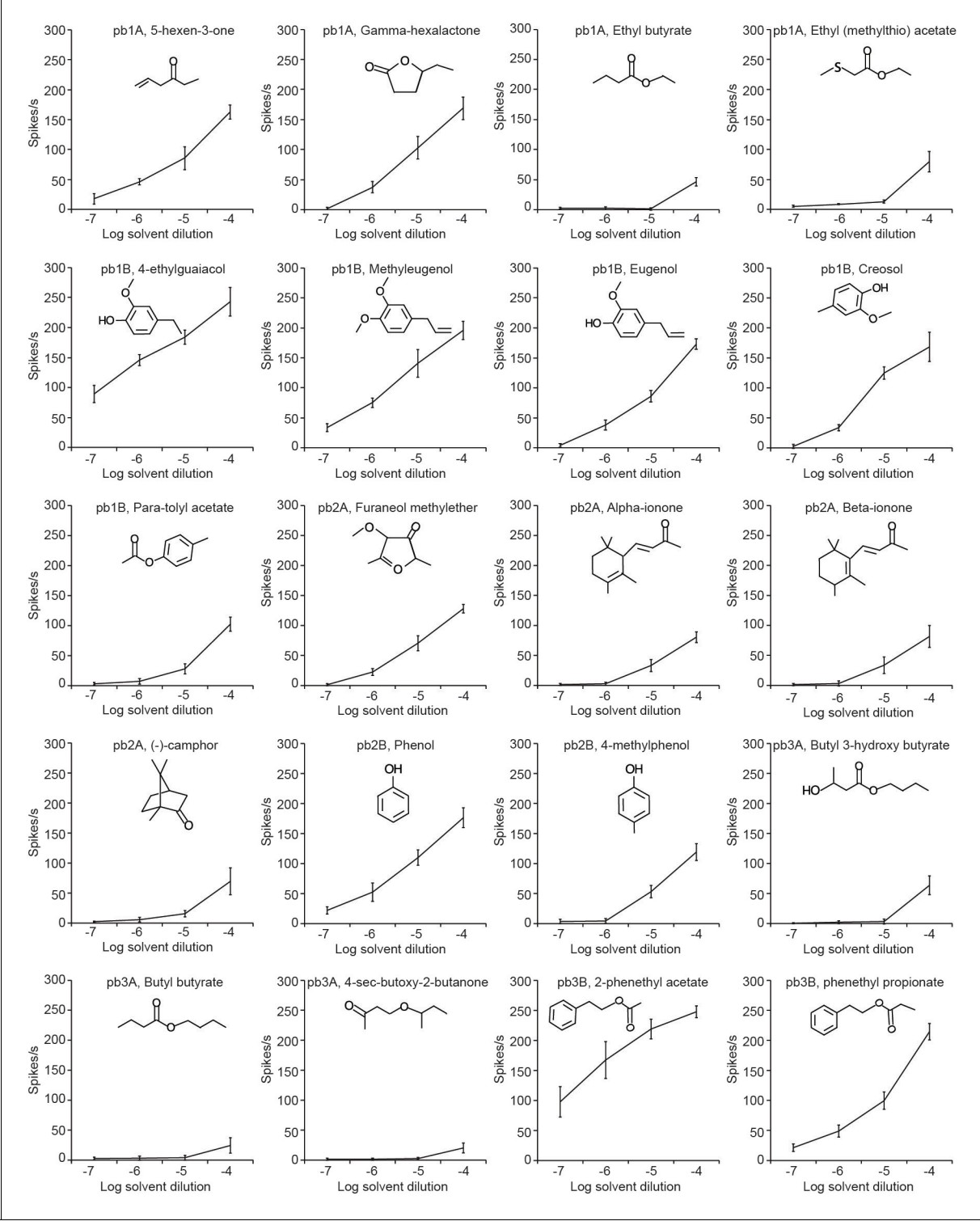

**Figure 2.** SSR dose-response curves for each MP-OSN stimulated with its physiologically active compounds (n = 5). Error bars represent SEM.

The following source data is available for figure 2:

**Source data 1.** Raw data for all dose-dependency curves presented in *Figure 2*.

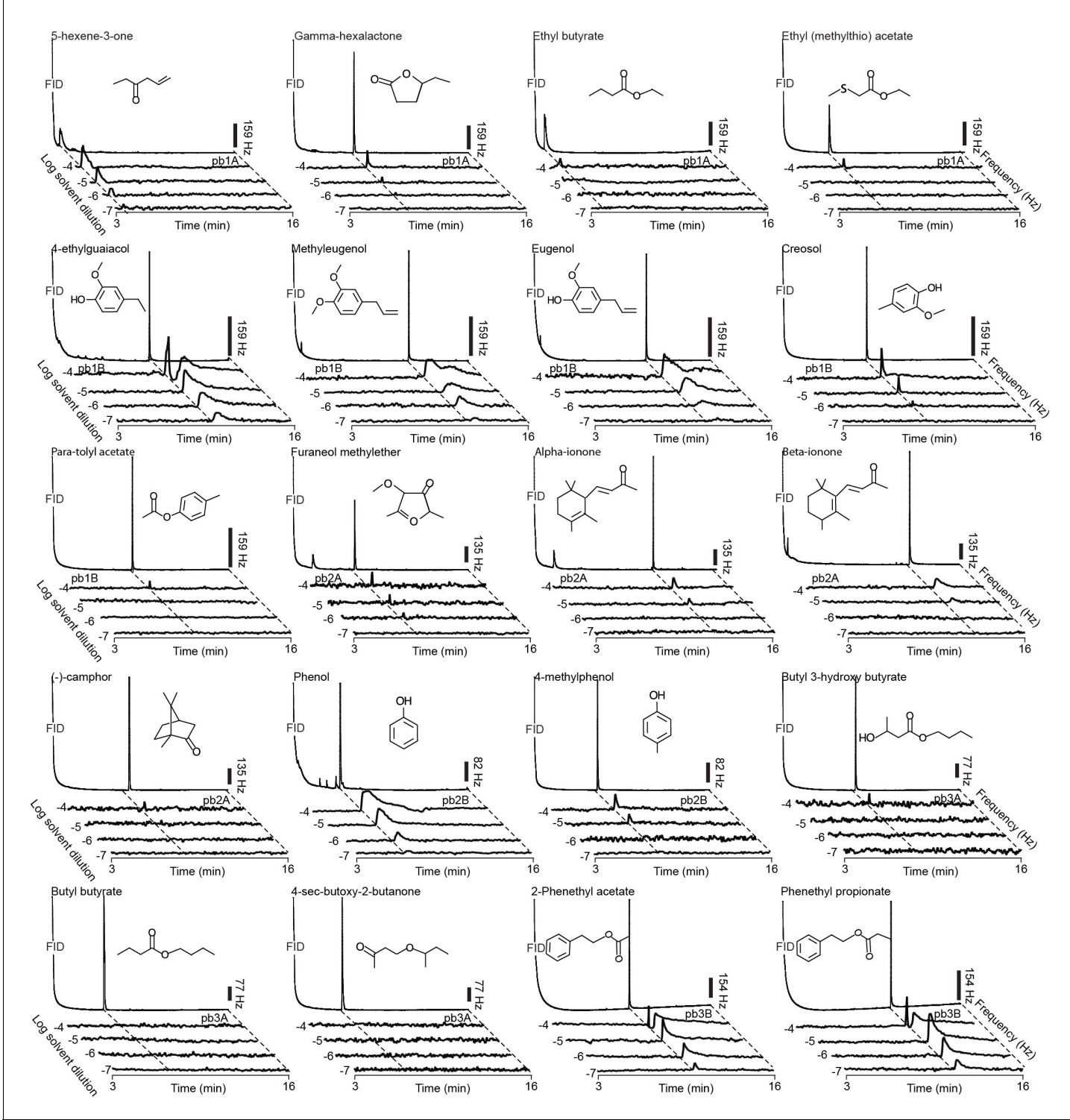

**Figure 3.** Representative GC-SSR dose-response traces for each MP-OSN stimulated with its physiologically active compounds (n = 3). Scale bars represent the neuronal firing rate [hz].

The following source data is available for figure 3:

**Source data 1.** Raw data for all GC-SSR results presented in *Figure 3*.

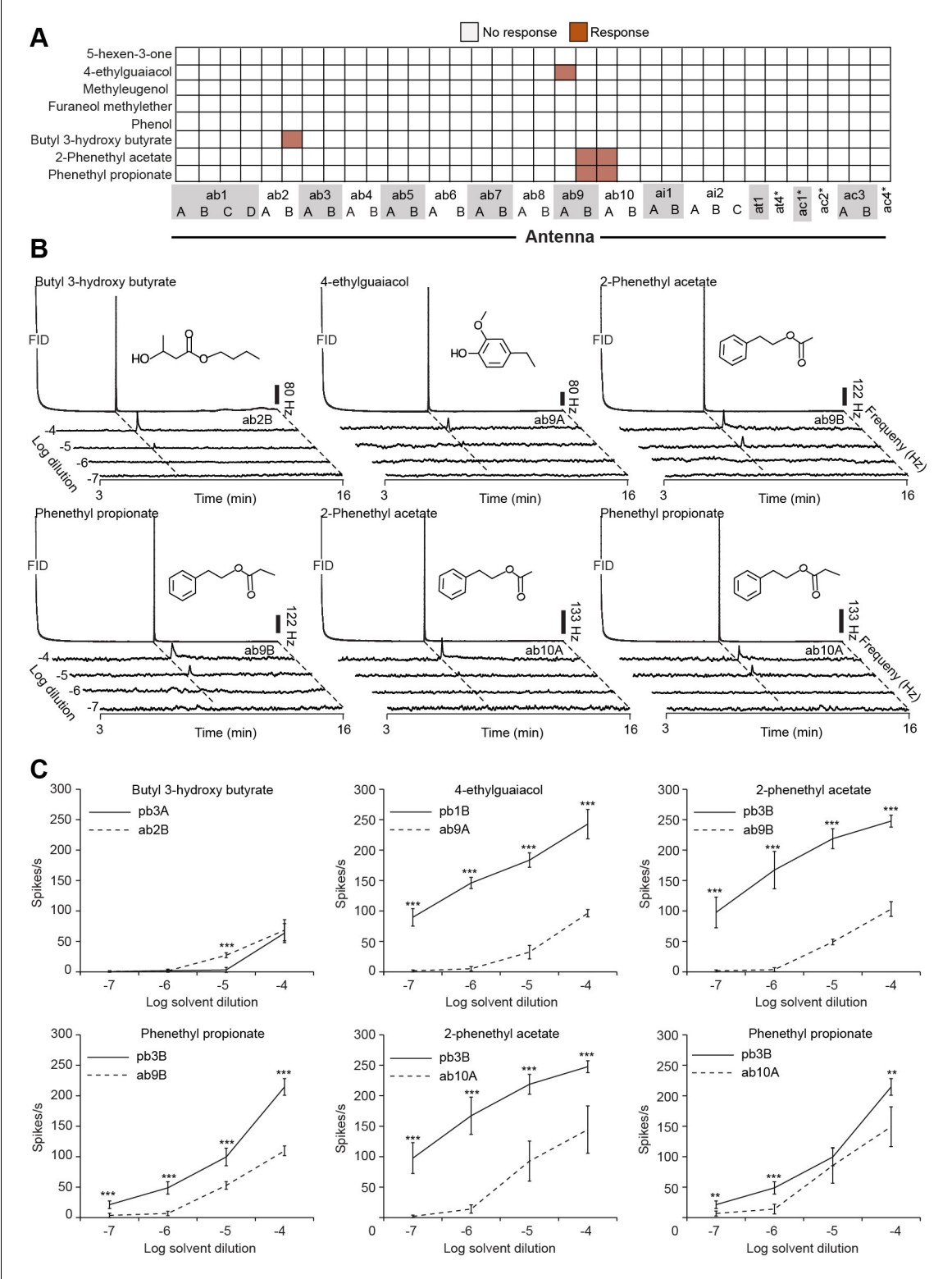

**Figure 4.** MP-OSNs are dedicated to detect specific chemical compounds. (**A**) Presence/Absence matrix of the GC-SSR responses of the MP-OSNs best activators across the Ant-OSNs (n = 3, dilution, $10^{-4}$ in hexane). Asterisks denote the total response of a sensillum type when spike sorting of OSNs failed. (**B**) Representative traces of GC-SSR dose response relationship from ab2A, ab9A, ab9B and ab10A OSNs (n = 3). Scale bars represent the neuronal firing rate [hz]. (**C**) SSR dose-response curves. Error bars represent SEM. The symbols ** and *** indicate statistically significant differences between OSN types with p<0.001, and p<0.0001, respectively (two-tailed Independent Samples T Test, n = 5).
*Figure 4 continued on next page*

*Figure 4 continued*

The following source data and figure supplement are available for figure 4:

**Source data 1.** Raw data of the comparison of antennal and palp OSN-responses to the best ligands of palp OSNs presented in *Figure 4*.
**Figure supplement 1.** Responses of Ant-OSNs to palp best activators.

(*Goldman et al., 2005*), in our PCA. (-)-fenchone distributed widely from furaneol methylether and instead clustered with the other three ligands of pb2A (*Figure 1E*, *Figure 1—source data 1*). This result suggests that the responsiveness of pb2A to furaneol methylether was due to the expression of Or33c, while the responsiveness of pb2A to (-)-camphor, alpha- and beta-ionone was due to the expression of Or85e.

**Table 2.** Best activators of MP-OSNs.

| Palp OSN | Odorant | Chemical structure | Detection threshold |
|---|---|---|---|
| pb1A | 5-hexen-3-one | | $10^{-7}$ |
| pb1B | 4-ethylguaiacol/Methyleugenol | | $10^{-7}$ |
| pb2A | Furaneolmethylether | | $10^{-6}$ |
| pb2B | Phenol | | $10^{-7}$ |
| pb3A | Butyl 3-hydroxy butyrate | | $10^{-4}$ |
| pb3B | 2-Phenethyl acetate/Phenethyl propionate | | $10^{-7}$ |

## Each MP-OSN is either the sole or the primary detector for specific chemical compounds

To determine which of the identified ligands is the best activator for each MP-OSN, we examined the dose-response relationships in SSR (*Figure 2*, *Figure 2—source data 1*) and GC-SSR (*Figure 3*, *Figure 3—source data 1*) experiments. Only one best ligand was identified for most MP-OSNs except for pb1B and pb3B, where two best ligands for each were identified (*Figure 2*). The detection threshold of pb1A, pb1B, pb2B and pb3B for their best activators was $10^{-7}$ dilution, whereas the detection threshold of pb2A and pb3A was $10^{-6}$ and $10^{-4}$ dilution, respectively (*Figure 2*). This high sensitivity suggests that the maxillary palps could be involved in evaluating odor sources over long distance similar to the antennae.

Several studies have suggested the existence of a labeled-line mode of odor coding in the olfactory and gustatory systems that signifies the presence of ecologically relevant signals of high biological importance (reviewed in *Depetris-Chauvin et al., 2015*). We next tested whether the best activators of MP-OSNs are detected via a single information channel. We screened all OSN types present on the antenna with these activators at $10^{-4}$ dilution using GC-SSR (*Figure 4A*, *Figure 4—figure supplement 1*, *Figure 4—source data 1*). We used this dosage because it is the maximum concentration that we can inject in the GC. Four of these best activators (5-hexen-3-one, methyleugenol, furaneol methylether, and phenol) elicited no response from any of the Ant-OSNs, while the other four triggered responses from four additional Ant-OSNs (*Figure 4A*, *Figure 4—figure supplement 1*). Interestingly, when odors activated both Ant-OSNs and MP-OSNs (4-ethylguaiacol: Or69a and Or71a; 2-phenethyl acetate and phenethyl propionate: Or67a and Or85d; butyl 3-hydroxy butyrate: Or59c and Or85a) in the latter two cases the receptor pairs cluster on a phylogenetic tree, suggesting a shared ancestor (*Robertson et al., 2003*).

To know the primary olfactory detector of the four activators that activate both MP- and Ant-OSNs, we performed dose-response relationships, but this time from the activated Ant-OSNs (*Figure 3B,C*). Three of the compounds were primarily detected by the MP-OSNs as the $10^{-5}$ detection threshold for these Ant-OSNs was two orders of magnitude higher than the $10^{-7}$ detection threshold for MP-OSNs. In addition, the number of spikes elicited by these three compounds at any tested concentration from MP-OSNs is significantly higher from Ant-OSNs except for phenethyl propionate at $10^{-5}$ concentration from pb3B and ab10A. The fourth activator, butyl 3-hydroxy butyrate, was primarily detected by the Ant-OSN ab2B. The detection threshold of ab2B to this compound was one order of magnitude ($10^{-5}$ dilution) lower than that of the corresponding MP-OSN pb3A ($10^{-4}$ dilution). Together, this data suggests that the MP-OSNs are either the sole or the primary detectors of ecologically relevant concentrations of 5-hexen-3-one, 4-ethylguaiacol, methyleugenol, furaneol methylether, phenol, 2-phenethyl acetate, and phenethyl propionate (*Table 2*).

## Contribution of MP-OSNs to short-range and long-range attraction

We next screened innate behavioral responses of flies to the best activators of MP-OSNs. We used trap and T-maze assays to measure short-range attraction, and wind tunnel assays to measure long-range attraction. In trap and T-maze experiments, we used $10^{-4}$ concentration, which is similar to the concentration used to measure the specificity of these ligands to different OSN types. In wind tunnel experiments, we used $10^{-2}$ concentration because the wind tunnel is supplied with a continuous airstream (0.3 m/s), which further dilutes this concentration. Six out of the eight tested compounds were behaviorally active; two compounds, 5-hexen-3-one and furaneol methylether, in T-maze assays, four compounds, 4-ethylguaiacol, methyleugenol, 2-phenethyl acetate and phenethyl propionate, in trap assays, and one compound, furaneol methylether, in wind tunnel assays (*Figure 5A*, *Figure 5—source data 1*). The finding that odors are differentially attractive in the trap and the T-maze assays is not new. E.g, the well-known *Drosophila* attractant, ethyl acetate, is attractive in T-maze assays (*Farhan et al., 2013*) and neutral in trap assays (*Knaden et al., 2012*). Part of the explanation of this variation might be due to flies flying in traps assays for 24 hr, while walking in T-maze assays for only 40 min. However, as so far never any odor was observed to be attractive in one and repellent in the other assay, we regard each odor that elicited at least attraction in one assay as attractive.

We next tested the behavioral responses of anosmic *Orco[2]* mutant flies, lacking the co-receptor necessary for the function of canonical Or receptors (*Larsson et al., 2004*), to the behaviorally active

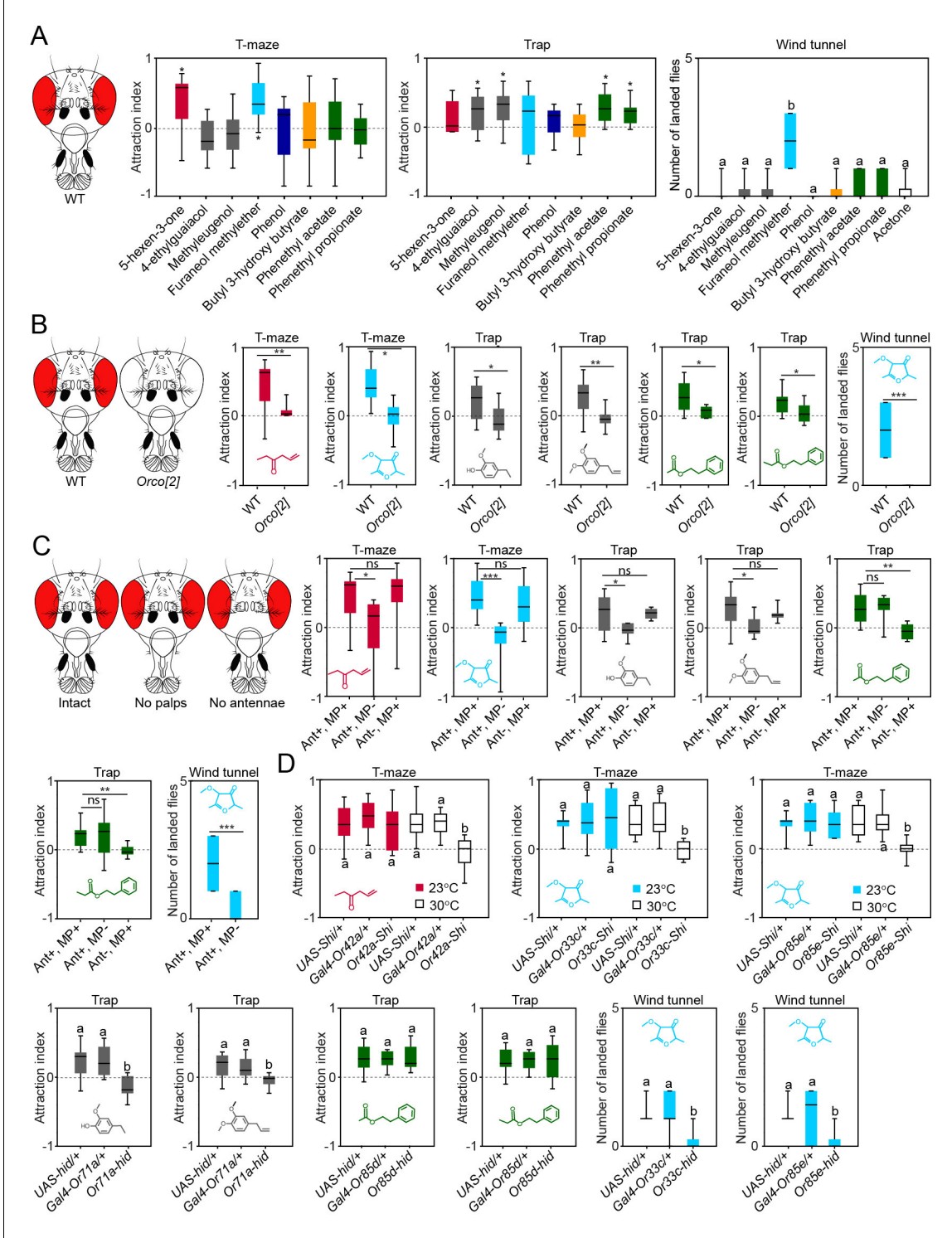

**Figure 5.** Contribution of the maxillary palp to the behaviors evoked by the palp best activators. (**A**) Behavioral responses of WT flies to the palp best activators (10⁻⁴ dilution used for trap and T-maze experiments, and 10⁻² dilution used for wind tunnel experiments). For T-maze and trap assays, the symbol * indicates significant differences from the solvent control (p<0.05, Wilcoxon signed rank test, n = 10). For wind tunnel assays, different letters indicate significant differences between groups (p<0.05, Kruskal Wallis test with Dunn's multiple comparison, n = 10). (**B**) Behavioral responses of WT and *Orco[2]* flies to the behaviorally active compounds (10⁻⁴ dilution used for trap and T-maze experiments and 10⁻² dilution used for wind tunnel experiments). The symbols *, ** and *** indicate statistically significant differences between the attraction indices of the genotypes with p<0.01, p<0.001, and p<0.0001, respectively (two-tailed Mann-Whitney U test, n = 10). (**C**) Behavioral responses of WT (Ant+, MP+), palp-amputated flies (Ant+, MP-) and antenna-amputated flies (Ant-, MP+) to the behaviorally active compounds (10⁻⁴ dilution used for trap and T-maze experiments and 10⁻²

*Figure 5 continued on next page*

*Figure 5 continued*

dilution used for wind tunnel experiments). The symbols *, ** and *** indicate statistically significant differences between groups with p<0.01, p<0.001, and p<0.0001, respectively; 'ns' indicates no significant differences between groups (Kruskal Wallis test with Dunn's multiple comparison for selected groups, n = 10). (D) Behavioral responses of flies with a killed or silenced specific MP-OSN population, the corresponding parental lines, and WT flies. Different letters indicate significant differences between groups (Kruskal Wallis test with Dunn's multiple comparison). Black line: median; boxes: upper and lower quartiles; whiskers: minimum and maximum values.

The following source data and figure supplements are available for figure 5:

**Source data 1.** Raw data of all behavioral experiments with *D. melanogaster* presented in *Figure 5*.

**Figure supplement 1.** Behavioural effects of palp odors on WT and *Orco[2]* flies.

**Figure supplement 1—source data 1.** Raw data of all behavioral experiments with *D. melanogaster* and palp activating odors at high concentration presented in *Figure 5—figure supplement 1*.

**Figure supplement 2.** Behavioral responses of male and female WT flies to 10% vinegar in wind tunnel experiments. 'ns' indicates no statistically significant differences between groups (p>0.05, two-tailed Mann-Whitney U Test, n = 10).

**Figure supplement 2—source data 1.** Raw data of wind-tunnel experiments performed with female and male *D. melanogaster* presented in *Figure 5—figure supplement 2*.

compounds. In contrast to WT flies, *Orco[2]* mutant flies were not attracted by these compounds (*Figure 5B*, *Figure 5—source data 1*), suggesting that the flies' behavior displayed to these compounds requires Or genes.

In the T-maze many pure chemicals become repellent at higher concentrations (*Strutz et al. 2014*). We, thus, measured innate responses of flies to $10^{-2}$ concentration of these ligands. We found, indeed, that flies are strongly repelled by 5-hexen-3-one, 4-ethylgauaicol, and phenol (*Figure 5—figure supplement 1* and *Figure 5—source data 1*). Interestingly, the aversion elicited by this concentration was independent of functioning Ors (*Figure 5—figure supplement 1* and *Figure 5—source data 1*). We, therefore, excluded this concentration in our further experiments. From our results we conclude that the palp best ligands represent positive cues at lower concentration, and that the processing of this information requires Or genes, while the processing of higher concentrations seems to be independent of functioning Ors.

To ensure that the behaviors evoked by the active compounds were mediated through the maxillary palps, we surgically removed either maxillary palps or antennae (we excluded wind tunnel experiments here, as the antenna has been shown to be involved in flight control [*Budick et al. 2007*]), and then tested behavioral responses of the manipulated flies. For 4-ethylguaiacol, 5-hexen-3-one, methyleugenol and furaneol methylether, amputation of the palps resulted in loss of attraction, while amputation of the antennae did not affect the behavior (*Figure 5C*, *Figure 5—source data 1*). Interestingly for 2-phenethyl acetate and phenethyl propionate amputation of the palps had no effect on the behavior, while amputation of the antennae abolished attraction elicited by these two compounds (*Figure 5C*, *Figure 5—source data 1*). We next killed or silenced a specific MP-OSN population using the temperature-sensitive mutant dynamin shibire^ts (*UAS-shi^ts*) or head involution defective (*UAS- hid*). We avoided using the *shibire^ts* effector in trap assays and wind tunnel experiments to avoid any temperature effect on flying flies. In T-maze and trap assays, the behavioral response to the corresponding ligand was abolished except for 2-phenethyl acetate and phenethyl propionate (*Figure 5D*, *Figure 5—source data 1*). In wind tunnel experiments, killing pb2A OSNs via expression of *hid* from Or33c- or Or85e-promoter significantly reduced attraction towards furaneol methylether compared to both parental lines (*Figure 5D*, *Figure 5—source data 1*). Taken all together, we conclude that the maxillary palp contains olfactory channels that mediate both short- and long-range attraction to specific chemical compounds.

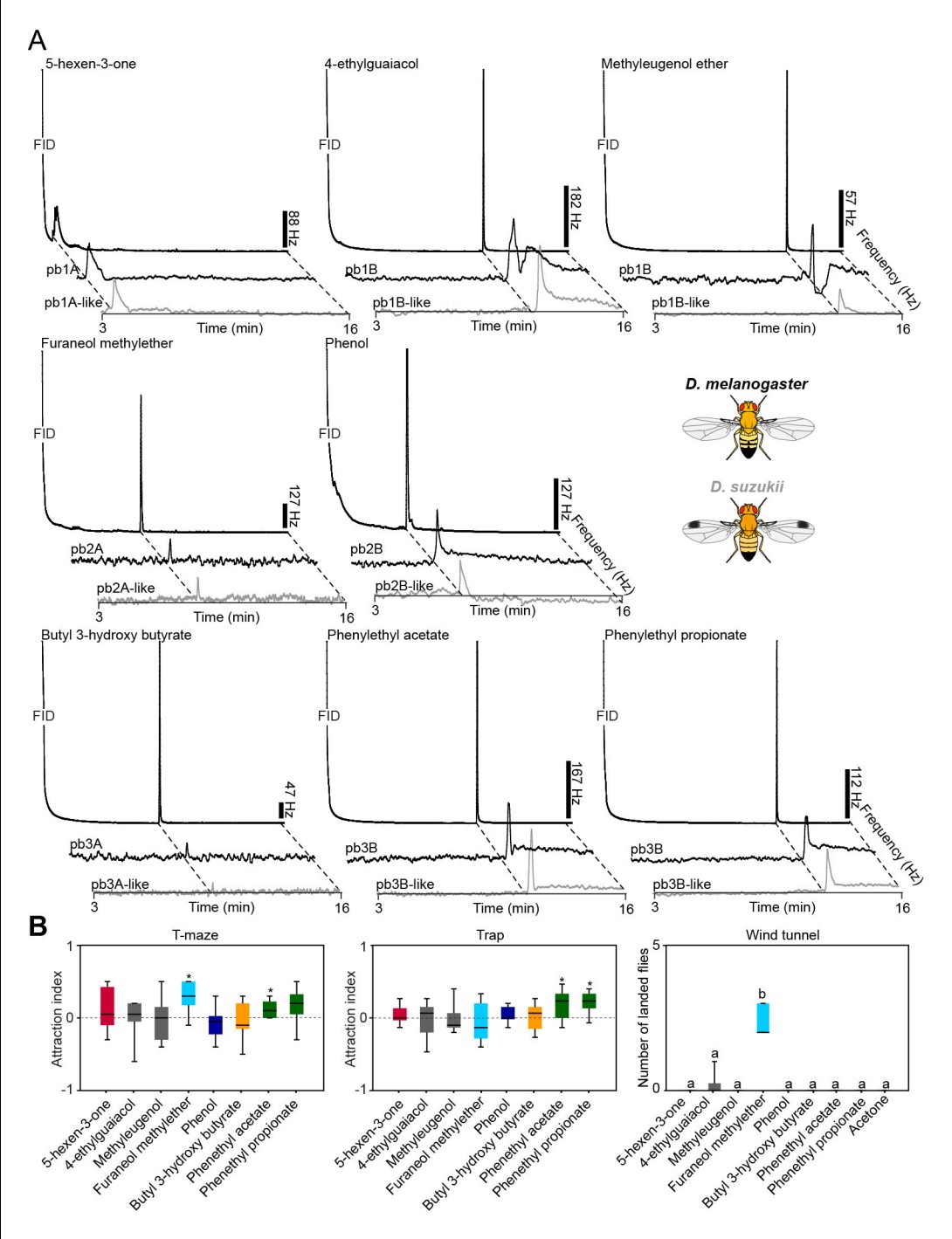

**Figure 6.** Organization, detection, and or genes of MP-OSNs are conserved in *D. suzukii*. (**A**) Representative GC-SSR traces from palp OSNs in *D. suzukii* and *D. melanogaster*, stimulated with the palp best ligands (dilution $10^{-4}$) (n = 3). (**B**) Behavioral responses of *D. suzukii* to *D. melanogaster* palp best activators ($10^{-4}$ dilution used for trap and T-maze assays, and $10^{-2}$ dilution used for wind tunnel experiments). For T-maze and trap assays, the symbol * indicates significant differences from the solvent control (p<0.05, Wilcoxon signed rank test, n = 10). For wind tunnel assays, different letters indicate statistically significant differences between groups (p<0.05, Kruskal Wallis with Dunn's multiple comparison). Black line: median; boxes: upper and lower quartiles; whiskers: minimum and maximum values.

The following source data is available for figure 6:

*Figure 6 continued on next page*

*Figure 6 continued*

**Source data 1.** Raw data of physiological and behavioral responses of *D. suzukii* presented in *Figure 6*.

## Organization, detection, and Or genes of MP-OSNs are conserved in D. suzukii

We selected *D. suzukii* to test whether the organization, detection and Or genes of MP-OSNs are conserved in another species. *D. suzukii* has recently invaded North America and Europe from Asia (*Rota-Stabelli et al., 2013*) and has become a serious agricultural pest for soft fruits causing devastating economic cost for farmers each year. Unlike most drosophilid flies including *D. melanogaster* that *D. suzukii* feed and oviposit on undamaged, ripening fruits. Thus, *D. suzukii* represents an interesting neuroethological model to study olfactory changes that parallel the evolutionary shift in the preference towards ripening over fermenting fruits.

We hence screened the MP-OSNs of *D. suzukii* with the best activators of *D. melanogaster* using GC-SSR. We found that the maxillary palp of *D. suzukii* contains three sensillum types as found in *D. melanogaster*, and that OSN types and their pairing within a particular sensillum type in *D. suzukii* are the same as in *D.* melanogaster. In addition, our screen revealed that the detection of these compounds is also conserved in *D. suzukii* (*Figure 6A*). Whether MP-OSNs of *D. suzukii* are the sole or the primary detectors of these ligands as found in *D. melanogaster* remains subject for future investigation.

We next aimed to know whether Or genes underlying these responses are also conserved in MP-OSNs of *D. suzukii*. We extracted the ortholog sequences of the genes expressed in the MP-OSNs in *D. melanogaster* from the public database of the *D. suzukii* genome (http://spottedwingflybase.oregonstate.edu/). These genes were then amplified from cDNA of our lab strain of *D. suzukii* (stock no. 14023–0311.01), cloned and sequenced. Five gene sequences were identified in full length, while the other two gene sequences (Or85d and Or85e) were partial. The gene sequences generated from the cDNA of our lab strain were submitted to the European Nucleotide Archive under the accession numbers LT555550-LT555555. We next aligned the amino acid sequences of the genes generated from the cDNA of our lab strain with those of *D. melanogaster* to compare their similarities. This comparison revealed that the amino acid sequences of these genes are well conserved in the *D. suzukii* genome: Or33c (82.2%), Or42a (90%), Or46a (83.5%), Or59c (78.4%), Or71a (81.4%), Or85d (86.5%) and Or85e (85%). We thus demonstrate that the organization, the detection of *D. melanogaster* palp best activators and the *D. melanogaster* Or genes of MP-OSNs are conserved in *D. suzukii*.

Like *D. suzukii* and *D. melanogaster*, also other closely related species share the same set of orthologs of olfactory genes expressed in MP-OSNs (*Guo and Kim, 2007*; *de Bruyne et al., 2010*), which might be either due to phylogenetic constraints or to their similar feeding habits (all examined species in this group feed on fruit-associated yeast). However, differences have been described for more distantly related species, such as *D. mojavensis* (cactus feeder), *D. virilis* (sap feeder), *D. grimshawi* (tree feeder) and *Scaptomyza flava* (leaf feeder). *D. mojavensis*, *D. virilis* and *D. grimshawi* have lost orthologs of Or59c, which is expressed in the *D. melanogaster* pb3A (*Guo and Kim, 2007*; *de Bruyne et al., 2010*), whereas *S. flava* has lost the ortholog of Or85d, which is expressed in the *D. melanogaster* pb3B (*Goldman-Huertas et al., 2015*). In line with these notions pb3A was not found during electrophysiological recordings from palp sensilla of *D. virilis* (*de Bruyne et al., 2010*).

While the MP-OSNs of all close relatives of *D. melanogaster* express the same set of olfactory receptors, the similar tuning of these MP-OSNs that we found in *D. suzukii* and *D. melanogaster* cannot necessarily be assumed for all species of the *D. melanogaster* species group. On the antenna e.g. the change of only few amino acids in a *D. sechellia* ortholog of the *D. melanogaster* Or22a gene has been shown to result in changed tuning curves of the corresponding OSN (*Dekker et al., 2005*). Hence, future studies will reveal whether the tuning of MP-OSNs is generally more conserved than the one of Ant-OSNs

## Conservation of the behavioral readouts to palp best activators in D. suzukii

To investigate whether the behavioral readouts of the olfactory inputs to the palp best activators are conserved in *D. suzukii*, we next examined innate responses of *D. suzukii* to these activators in trap, T-maze and wind tunnel assays (*Figure 6B*). The yeast producing volatile 2-phenethyl acetate and phenethyl propionate (*Dweck et al., 2015*; *Christiaens et al., 2014*), elicited positive chemotaxis in trap and/or T-maze two-choice assays, while the ripening signal, furaneol methylether (*Ulrich et al., 1997*), induced positive chemotaxis and upwind attraction. The fermentation signals produced by the metabolism of hydrocinnamic acids in fruits by yeasts, 4-ethylguaiacol and methyleugenol (*Dweck et al., 2015*), did not induce any behavioral response in *D.suzukii* in contrast to *D. melanogater*. *Krause Pham and Ray (2015)* reported a similar case, where they found that the avoidance behavior of *D. melanogaster* to $CO_2$, which is highly emitted by ripe fruits, is not conserved any more in *D. suzukii*, although the $CO_2$ detection and the genes responsible for this detection are conserved.

## Conclusions

In summary, we demonstrate that the maxillary palp in the vinegar fly, *D. melanogaster* contains OSNs that mediate both short-and long-range attraction evoked by specific chemical compounds in the flies' ecological niche. Interestingly, although the sensitivity of MP-OSNs was described to be rather low (*de Bruyne et al., 1999*; *Hallem and Carlson, 2006*), which led to the assumption that MP-OSNs are basically involved in taste enhancement (*Shiraiwa et al., 2008*), we show that their sensitivity to some compounds can be as high as in Ant-OSNs. Furthermore, MP-OSN specific ligands did not only attract walking flies over short distance, but in one case (furaneol methylether, Or33c) even motivated flies to fly towards the source.

Finally we found that although the detection of *D. melanogaster* palp best activators and Or genes of MP-OSNs are conserved in the agricultural pest *D. suzukii*. However, only behavioral readouts to 2-phenethyl acetate and phenethyl propionate produced by yeast volatiles, and furaneol methylether that represent ripening signal in strawberries are conserved. Contrary, behavioral readouts to the yeast metabolites 4-ethylguaiacol and methyleugenol that represent fermentation signals, are not conserved in this pest species These behavioral changes might represent a taxon-specific adaptation to the newly emerging ecological niche of this pest species.

# Materials and methods

## Fly stocks

All experiments with wild type (WT) *D. melanogaster* were carried out with the Canton-S strain. *D. suzukii* (stock no. 14023–0311.01) was obtained from the UCSD *Drosophila* Stock Center (www.stockcenter.ucsd.edu). Transgenic lines were obtained from the Bloomington *Drosophila* stock center (http://flystocks.bio.indiana.edu/), except for *UAS-Shibire*[ts], which was a kind gift from G.M. Rubin (Janelia Farm Research Campus, USA).

## Complete genotypes of all strains used in this study

Shi[ts]-Or42a
 Females
 $w^{1118}/w^{1118}$; +/+; UAS-Shi[ts]/+
 $w^{1118}/w^{1118}$; +/+; +/Or42a-Gal4
 $w^{1118}/w^{1118}$; +/+; UAS-Shi[ts]/Or42a-Gal4
 Shi[ts]-Or33c
 Females
 $w^{1118}/w^{1118}$; +/+; UAS-Shi[ts]/+
 $w^{1118}/w^{1118}$; +/+; +/Or33c-Gal4
 $w^{1118}/w^{1118}$; +/+; UAS-Shi[ts]/Or33c-Gal4
 Shi[ts]-Or85e
 Females
 $w^{1118}/w^{1118}$; +/+; UAS-Shi[ts]/+
 $w^{1118}/w^{1118}$; +/+; +/Or85e-Gal4

w[1118]/w[1118]; +/+; UAS-Shi[ts]/Or85e-Gal4
hid-Or71a
Females
w[1118]/w[1118]; UAS-hid/+; +/+
w[1118]/w[1118]; +/+; +/Or71a-Gal4
w[1118]/w[1118]; UAS-hid/+; +/Or71a-Gal4
Males
w[1118]/Y; UAS-hid /+; +/+
w[1118]/Y; +/+; +/Or71a-Gal4
w[1118]/Y; UAS-hid/+; +/Or71a-Gal4
hid-Or85d
Females
w[1118]/w[1118]; UAS-hid/+; +/+
w[1118]/w[1118]; +/+; +/Or85d-Gal4
w[1118]/w[1118]; UAS-hid/+; +/Or85d-Gal4
Males
w[1118]/Y; UAS-hid /+; +/+
w[1118]/Y; +/+; +/Or85d-Gal4
w[1118]/Y; UAS-hid/+; +/Or85d-Gal4

## Odor samples

Fruit samples were either ripe or in early fermentation stage. Microorganisms were kept on strain-specific media (HiMedia, http://www.himedialabs.com), following standard protocols. Mammalian fecal samples were provided by the Leipzig Zoo.

## Headspace collections

The headspaces of the different samples were collected for 24 hr on a Super-Q filter (50 mg, Analytical Research Systems, Inc., www.ars-fla.com). The samples were placed individually in an l liter laboratory glass bottle that was halfway filled with samples and closed with a custom-made polyether ether ketone (PEEK) stopper. Airflow at 0.5 L/min was drawn through the flask by a pressure pump. Filters were eluted with 300 µl hexane and samples stored at -20°C until analysis.

## SSR/GC-SSR/GC-MS

Adult flies were immobilized in pipette tips, and the palps or antennae were placed in a stable position onto a glass coverslip. Sensilla were localized under a binocular at 1000× magnification, and the extracellular signals originating from the OSNs were measured by inserting a tungsten wire electrode into the base of a sensillum. The reference electrode was inserted into the eye. Signals were amplified (10×; Syntech Universal AC/DC Probe; www.syntech.nl), sampled (10,667 samples/s), and filtered (100–3000 Hz with 50/60-Hz suppression) via a USBIDAC connection to a computer (Syntech). Action potentials were extracted using Syntech Auto Spike 32 software. For SSR, neuron activities were recorded for 10 s, starting 2 s before a stimulation period of 0.5 s. Responses from individual neurons were calculated as the increase (decrease) in the action potential frequency (spikes/s) relative to the prestimulus frequency. For GC-SSR, neuron activities were recorded for 1220 s, the time of a single GC run. For GC stimulation, 1 µl of the odor sample was injected into a GC (Agilent 6890, column: DB5, 30 m long, 0.32 mm id, 0.25 µm film thickness; inlet at 250°C, oven: 50°C for 2 min, then 15°C min[-1] up to 250°C, held for 5 min; carrier gas: helium, 2.0 ml min[-1] constant flow). The GC was equipped with a 4-arm effluent splitter (Gerstel, www.gerstel.com), with split ratio 1:1 and N2(30.3 kPa) as makeup gas. One arm was connected with the flame ionization detector (FID) of the GC and the other arm introduced into a humidified air stream (200 ml min[-1]). GC-separated components were directed toward the palps of the mounted fly. Signals from OSNs and FID were recorded simultaneously. Headspace samples were analyzed by GC-MS (Agilent 6890GC & 5975bMS, Agilent Technologies, www.agilent.com).

## Chemicals

All odorants tested were purchased from commercial sources (Sigma, http://www.sigma-aldrich.com and TCI America, http://www.tcichemicals.com/en/us/) except for 5-hexen-3-one and butyl 3-hydroxy butyrate, which were synthesized in house from propionitrile and allyl bromide using the protocol of *Rousseau et al. (1981)*, and from ethyl-3-oxobutanoate using the protocol of *Padhi et al. (2003)*, respectively.

## Trap experiments

In this paradigm thirty 4–5 day-old mated female flies that were starved with free access to water for 24 hr were introduced into a small box (length, 10 cm; width, 8 cm; height, 10 cm) that contained two smaller containers (height, 4.5 cm; diameter, 3 cm). The reason for performing these experiments with only females is that hungry females live longer than hungry males. For 24 hr, flies could enter (but not leave) these containers through a pipette tip (tip opening, 2 mm). Containers were equipped with the lid of an Eppendorff cap that was loaded with either an odorant or solvent. The attraction index (AI) was calculated as AI = (O−C) / 30, where O is the number of flies entered the odorant containing trap and C is the number of flies entered the solvent containing trap. The index could range from -1 (complete avoidance) to 1 (complete attraction). A value of 0 characterizes no response, i.e. the odor is not detected or is neutral. Experiments were carried out in a climate chamber at 20, 25 or 30°C and 70% humidity. Experiments were started in the morning with 12 hr of white light, followed by 12 hr of no light.

## T-maze experiments

T-maze experiments were carried out as described in *Stensmyr et al. (2012)*. In brief, thirty 4–5 days old starved and mated female and male flies were introduced into the bottom part of a t--shaped tube (length of each arm, 4 cm; diameter, 1 cm) and during 40 min were allowed to enter (but not to leave) via pipette tips (tip opening, 2 mm) eppendorff caps attached to the two upper arms of the t-shaped tube. The lids of the Eppendorff caps contained 0.5 ml agar (1%) that was loaded either with 50 μl of the odorant or with solvent only. The positions of odorant and solvent within the t-mazes were changed repeatedly. The attraction index (AI) was calculated as AI = (O-C)/ 30, where O is the number of flies entered the odorant containing trap and C is the number of flies entered the solvent containing trap. Experiments were carried out in a climate chamber at 20, 25 or 30°C and 70% humidity. For shibire experiments, flies were warmed at 30°C for 30 min prior to behavioral assays. All t-maze assays were performed under white light.

## Wind tunnel experiments

Free-flight experiments were performed in a wind tunnel that was built as described previously (*Becher et al., 2010*), with the airstream in the tunnel (0.3 m/s) produced by a fan and filtered through activated charcoal. The wind tunnel was maintained within a climate chamber set to 27°C and 50–55% humidity. Five flies (following suggestions from *Becher et al. (2010)* for highest responses rates in wind tunnel assays, we used 2 days old flies that were mated and starved for 24 hr) were together released at the center position of the downwind side of the tunnel. No differences between sexes were noted, and thus the data were pooled (*Figure 5—figure supplement 2*, *Figure 5—figure supplement 2—source data 1*). 50 μl of a $10^{-2}$ dilution of the odorant in acetone (solvent) was delivered onto a filter paper, which was placed in a plastic tube (diameter, 3 cm). The tube was horizontally suspended within the airstream in the center position of the upwind side of the tunnel. Flies landing at the tube were counted for the first 10 min after release. All wind tunnel experiments were performed under white light.

## Gene annotation

Annotated genomic sequences of *Drosophila suzukii* were obtained from SpottedWingFlyBase (http://spottedwingflybase.oregonstate.edu). *Drosophila melanogaster* Or sequences were downloaded from flybase.org. Using the BLAST algorithm we identified gene models of the *Drosophila melanogaster* orthologs of Or33c, Or42a, Or46a, Or59c, Or71a, Or85d and Or85e in the *Drosophila suzukii* genome. The gene models were curated manually by comparison with the sequences of *Drosophila melanogaster*. For gene sequences see *Supplementary file 1*.

## RT-PCR and cDNA cloning

The third antennal segment and palps of ~100 *D. suzukii* flies were collected and transferred to Eppendorf cups chilled on dry ice. Subsequently they were homogenized with ceramic beads for 15 min at 50 Hz in a TissueLyser LT (Qiagen, Hilden Germany). Total RNA was isolated using TRizol isolation following the manufacture´s protocol. The extracted total RNA was dissolved in RNAse free water. The quality was checked by gel electrophoresis and the concentration was measured photometrically. cDNA synthesis for RT-PCR was done by using SuperScript III First-Strand Synthesis Kit (Invitrogen, Life Technology, Grand Island, USA). RT-PCR was performed according to standard protocols, using primers of the table below with an annealing temperature of 57°C. PCR products were cloned into pCR2.1 vector (Invitrogen, Life Technology, Grand Island, USA). Sequencing was performed by Eurofins Genomics.

| Gene | Forward primer | Reverse primer |
|------|---------------|----------------|
| DsuzOr33c | 5´ACC ATG GTC ATC ATC GAT AGT GTT CAT 3´ | 5´CTA TAT ACC TTT CAC CCG CAC CAC 3´ |
| DsuzOr42a | 5´ATG GAG CTG CAA AGA ATC ATT CCG 3´ | 5´TCA ATC GTC TTC ATC AGA TTT GGC TAA 3´ |
| DsuzOr46a | 5´ACC ATG AGC AAC AGA GTG GAA ATC 3´ | 5´CTA ACT GTT GAC CCG CTT TAG CAA 3´ |
| DsuzOr59c | 5´ACC ATG AAG AAG CCG CTC TTT GAA CGT 3´ | 5´TTA GGG CTC TAC TTC CCC TGC ATT 3´ |
| DsuzOr71a | 5´ACC ATG GAT TAC GAC CGA ATT CGA CCA 3´ | 5´CTA TTG GTT CAT GTT GAG CAG CAA G 3´ |
| DsuzOr85d | 5´ATG GCA GCG AAG AAG CAA ACT CAA 3´ | 5´ TCA GGT ACT ATA CAT TGT GCG CAG 3´ |
| DsuzOr85e | 5´ATG GCC AGT CTT CAG TTC CAC GG 3´ | 5´GGG CGT GTT TCC ACCATG AGC 3´ |

## Data analysis

Chemometric analysis was performed as outlined in *Haddad et al. (2008)*, and was used as basis for a Principal Component Analysis (PCA) performed in PAST (folk.uio.no/ohammer/past/). Normality and homogeneity of variances were tested in SPSS (www.spss.com) to select appprociate statistical tests. All statistical tests were performed with SPSS or Graphpad Instat.

## Acknowledgements

We wish to thank Richard A Fandino and Jan E Bello for comments on the manuscript.

## Additional information

### Competing interests

BSH: Vice President of the Max Planck Society, one of the three founding funders of eLife, and a member of eLife's Board of Directors. The other authors declare that no competing interests exist.

### Funding

| Funder | Author |
|--------|--------|
| Max-Planck-Gesellschaft | Bill S Hansson |

The funders had no role in study design, data collection and interpretation, or the decision to submit the work for publication.

### Author contributions

HKMD, Final approval of submitted version, Conception and design, Acquisition of data, Analysis and interpretation of data, Drafting or revising the article; SAME, MAK, CK, AF, RS, Final approval of submitted version, Acquisition of data, Drafting or revising the article; JW, Final approval of submitted version, Analysis and interpretation of data, Drafting or revising the article, Contributed unpublished essential data or reagents; AS, Final approval of submitted version, Analysis and interpretation of data, Contributed unpublished essential data or reagents; EG-W, Final approval of

submitted version, Acquisition of data, Analysis and interpretation of data, Drafting or revising the article; MK, BSH, Final approval of submitted version, Conception and design, Analysis and interpretation of data, Drafting or revising the article

**Author ORCIDs**
Markus Knaden, http://orcid.org/0000-0002-6710-1071

## Additional files

**Supplementary files**
• Supplementary file 1. Gene sequences.

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
