## [Decision Letter]

Thank you for submitting your article "Olfactory Channels Associated with the *Drosophila* Maxillary Palp Mediate Short- and Long-range Attraction" for consideration by *eLife*. Your article has been favorably evaluated by a Senior editor and three reviewers, one of whom is a member of our Board of Reviewing Editors.

The reviewers have discussed the reviews with one another and the Reviewing Editor has drafted this decision to help you prepare a revised submission.

Summary:

This manuscript examines the ligand selectivity and function of the maxillary palp in olfactory coding in *Drosophila melanogaster* and *suzukii*. The authors use single sensilla electrophysiological recordings and GC-MS to isolate odors that activate the different classes of palp neurons in *melanogaster*, determine the sensitivity of the response, and compare with antennal sensitivity. They show that these odors influence short- and long- range olfactory behaviors and require the maxillary palp. Moreover, *suzukii* palp responds to the same odors. Overall, this is a potentially important analysis of olfactory responses in the maxillary palp and the role of these odors in behavior. However, there are major concerns about the methodology and documentation of the study that must be addressed.

1) The odorant concentrations used for the behavioral assays are far above those used for specific pb neuron activation. What odor concentrations are experienced by the fly at choice points? Do the odor concentrations used for behavioral experiments elicit the same activation patterns as the concentrations used for recording? The authors need to more definitely demonstrate that their behavioral conditions replicate their SSR data. As it is now, it is possible that the odorant concentrations they used may very well activate non-pb neurons, and it is that activation of these other olfactory neurons that are guiding the observed behaviors.

2) The authors should provide full SSR data (MP and antenna responses to all odors tested and all dilutions), raw or as plots similar to Figure 2, or as a table like Figure 3, and grade the responses based on intensities, for both antenna and MP responses; doing this will be necessary to assess olfactory specificity of MP-OSNs. Reviewers and authors will be familiar with the impact of e.g. the Hallem and Carlson 2006 on study of 24 antennal receptors, which included a table summarizing all odorant/receptor responses. 10 years later we should expect something similar for the 6 receptor neurons/7 ORs of the palp.

3) The behavioral results that are used to argue for short- and long-range attraction are not straightforward. For some odors, the authors observe attraction in T-maze but not in the trap assay, nor wind-assay, for others only in the trap-assay. The authors give no explanation for this variation and do not justify their subsequent choice only to consider those odor-assay combinations that gave positive results. Similarly, the authors provide no information on why in some assays only females were used, why sometimes flies were starved, or why the assay duration and fly age was varied between assays. The behavioral studies need to be better documented. Details required:

A) Odorant concentrations experienced by the fly (as mentioned above).

B) Genetic background of experimental animals. Are genetic reagents backcrossed (i.e., OrX-GAL4, and UAS-shibire) into the same genetic background?

C) Experimental conditions: were flies starved or fed? Were experiments performed in the dark or light? Age of flies? Virgins?

D) For significance, which 'controls' are the experimental groups being compared to? For example, in Figure 1, is the WT 30C flies compared to the Or42a-Shi 30C flies? This needs to be described more fully.

E) Did the authors perform positive controls to verify that their assays could identify repulsive behaviors? If the authors want to claim certain odorants guide attraction, they need to verify that they would have seen the lack of attraction (repulsion) in their assay if this was a possibility. Cyclopentanone has been found to be repellant to flies (Tauxe, 2013) as well as many other odorants.

F) T-maze assay. a) The authors use 40 minutes in the T-maze assay. Since the T-maze is used as a read-out for innate olfactory valence responses, shorter times should be used (2-5 minutes should be sufficient). b) Did the authors change the side in the T-maze that contained the odor tube?

4) The same wild-type data is apparently used repeatedly in Figure 4 in comparisons to different experimental condition. For example, WT bar representing 5-hexen-3-one is the same in 4A, 4B, 4C; WT bar representing strawberry furanone is the same in 4A, 4B, 4C. How did the authors control for changes in behavioral effects that often occur when control and experimental behaviors are not performed in parallel? Even slight variations in temperature and humidity can have large effects on olfactory behaviors.

[Editors' note: further revisions were requested prior to acceptance, as described below.]

Thank you for resubmitting your work entitled "Olfactory Channels Associated with the *Drosophila* Maxillary Palp Mediate Short- and Long-range Attraction" for further consideration at *eLife*. Your revised article has been favorably evaluated by a Senior editor, a Reviewing editor, and one reviewer.

The manuscript has been improved but there are some remaining text and figure clarification issues that need to be addressed before acceptance, as outlined below:

1) The GC-SSR axis in Figure 3 and Figure 4 is not clear. What is the scale? Should this not be represented as a vertical scale bar?

2) Figure 1. The main graphic (panel B) shows what odorants are present in different extracts. What is the connection to the response properties of the palp neurons? Rather than odor present, shouldn't neuron response be mapped?

3) Figure 4. What odor concentrations do these represent?

4) Figure 5. Controls should have been compared to solvent odor controls, and not to zero. For example, maybe in their assay, the solvent itself leads to some attraction. Or if given a choice between two solvent background odors, the behavior would become highly variable (larger spread) such that the significantly different from zero data points would no longer be significant.

5) Figure 1. Unknown spelled Unknow in pb1B section.

6) Genotypes, subsection “Complete genotypes of all strains used in this study”. w^1118^ instead of w^118^.

---

## [Author Response]

1) The odorant concentrations used for the behavioral assays are far above those used for specific pb neuron activation. What odor concentrations are experienced by the fly at choice points? Do the odor concentrations used for behavioral experiments elicit the same activation patterns as the concentrations used for recording? The authors need to more definitely demonstrate that their behavioral conditions replicate their SSR data. As it is now, it is possible that the odorant concentrations they used may very well activate non-pb neurons, and it is that activation of these other olfactory neurons that are guiding the observed behaviors.

We adjusted this in the text and wrote that "In trap and T-maze experiments, we used 10^-4^ concentration, which is similar to the concentration used to measure the specificity of these ligands to different OSN types. In wind tunnel experiments, we used 10^-2^ concentration because the wind tunnel is supplied with continuous airstream (0.3 m/s), which further dilutes this concentration.”

However, even when we assume that the used concentration activated additional OSNs, the manipulation of palp and antenna, and specific OSNs clearly showed the role of MP-OSNs in the observed behavior.

2) The authors should provide full SSR data (MP and antenna responses to all odors tested and all dilutions), raw or as plots similar to Figure 2, or as a table like Figure 3, and grade the responses based on intensities, for both antenna and MP responses; doing this will be necessary to assess olfactory specificity of MP-OSNs. Reviewers and authors will be familiar with the impact of e.g. the Hallem and Carlson 2006 on study of 24 antennal receptors, which included a table summarizing all odorant/receptor responses. 10 years later we should expect something similar for the 6 receptor neurons/7 ORs of the palp.

We provided the full SSR data and added Figure 1—figure supplement 1–Figure 1—figure supplement 8, Figure 2, [Supplementary-material SD2-data], [Supplementary-material SD3-data], [Supplementary-material SD4-data], [Supplementary-material SD4-data], and [Supplementary-material SD8-data].

3) The behavioral results that are used to argue for short- and long-range attraction are not straightforward. For some odors, the authors observe attraction in T-maze but not in the trap assay, nor wind-assay, for others only in the trap-assay. The authors give no explanation for this variation and do not justify their subsequent choice only to consider those odor-assay combinations that gave positive results. Similarly, the authors provide no information on why in some assays only females were used, why sometimes flies were starved, or why the assay duration and fly age was varied between assays. The behavioral studies need to be better documented. Details required:

We agree that it is a bit confusing that flies become attracted to one odor within one but not within another assay. However, it has been shown many times that, whether a fly becomes attracted or not by an odorant strongly depends on the bioassay. We therefore decided to use several bioassays. We agree that for the most attractive odorant one would assume that the fly becomes attracted in all assays. However, as already shown even for the well-known attractant ethyl acetate, some assays fail to show attraction (ethyl acetate: Knaden et al. 2012). We, therefore regard an odor as attractive if it at least provokes attraction in one of the assays used. We added “The finding that odors are differentially attractive in the trap and the T-maze assays is not new. E.g., the well-known *Drosophila* attractant, ethyl acetate, is attractive in T-maze assays (Farhan et al., 2013) and neutral in trap assays (Knaden et al., 2012). Part of the explanation of this variation might be due to flies flying in traps assays for 24 h, while walking in T-maze assays for only 40 min. However, as so far never any odor was observed to be attractive in one and repellent in the other assay, we regard each odor that elicited at least attraction in one assay as attractive.”

A) Odorant concentrations experienced by the fly (as mentioned above).

See our arguments above.

B) Genetic background of experimental animals. Are genetic reagents backcrossed (i.e., OrX-GAL4, and UAS-shibire) into the same genetic background?

Yes. We added to the Materials and methods section this information and now give the complete genotypes of all strains used in this study.

C) Experimental conditions: were flies starved or fed? Were experiments performed in the dark or light? Age of flies? Virgins?

We apologize and have now added all information on mating state, age, and experimental conditions to the Materials and methods part.

*D) For significance, which 'controls' are the experimental groups being compared to? For example, in Figure 1, is the WT 30*°*C flies compared to the Or42a-Shi 30C flies? This needs to be described more fully.*

We agree that the comparison with WT flies is not correct. We have now deleted the WT data from this part of the figure and e.g. the Or42a-Shi 30C flies both with Or42a-Shi 23C flies as well as with the parental lines.

E) Did the authors perform positive controls to verify that their assays could identify repulsive behaviors? If the authors want to claim certain odorants guide attraction, they need to verify that they would have seen the lack of attraction (repulsion) in their assay if this was a possibility. Cyclopentanone has been found to be repellant to flies (Tauxe, 2013) as well as many other odorants.

We agree that such a positive control would be helpful and have added data from experiments with higher concentrations: “Many pure chemicals become repellent at higher concentrations (Strutz et al. 2014). We, thus, measured innate responses of flies to 10^-2^ concentration of these ligands. We found, indeed, that flies are strongly repelled by 5-hexen-3-one, 4-ethylgauaicol, and phenol (Figure 5—figure supplement 1 and Figure 5—source data 2). Interestingly, the aversion elicited by this concentration was independent of functioning Ors (Figure 5—figure supplement 1 and Figure 5—source data 2). We, therefore, excluded this concentration in our further experiments.”

F) T-maze assay. a) The authors use 40 minutes in the T-maze assay. Since the T-maze is used as a read-out for innate olfactory valence responses, shorter times should be used (2-5 minutes should be sufficient).

Different T-maze assays exist. In the T-maze that people e.g. use for learning assays indeed flies need only 2-5 minutes for decision. However, the T-maze used in our manuscript was developed in the Sidiqi lab (used e.g. by Stensmyr et al. Cell 2012; Farhan et al. Scientific Reports 2013). As the flies need to pass a pipette tip to enter each arm of the T-maze, decision time needs to be longer.

b) Did the authors change the side in the T-maze that contained the odor tube?

Yes. We added this information to the Methods section.

4) The same wild-type data is apparently used repeatedly in Figure 4 in comparisons to different experimental condition. For example, WT bar representing 5-hexen-3-one is the same in 4A, 4B, 4C; WT bar representing strawberry furanone is the same in 4A, 4B, 4C. How did the authors control for changes in behavioral effects that often occur when control and experimental behaviors are not performed in parallel? Even slight variations in temperature and humidity can have large effects on olfactory behaviors.

As we now compare all genetic modified flies with their parental lines WT is not used as a control as often anymore (only to compare with flies with removed antennae or palps, and *Orco* flies). The condition inside the climatic chamber is computer controlled, which hopefully reduces variation between experiments. However, in addition, we did not perform all 10 replicates of a single experiment at the same time, but tested e.g. some WT replicates, some *Orco* replicates and some manipulated flies in parallel.

[Editors' note: further revisions were requested prior to acceptance, as described below.]

The manuscript has been improved but there are some remaining text and figure clarification issues that need to be addressed before acceptance, as outlined below:

1) The GC-SSR axis in Figure 3 and Figure 4 is not clear. What is the scale? Should this not be represented as a vertical scale bar?

We apologize for this and have now added a scale bar to both figures. We also mention scale bar in the figure legend.

2) Figure 1. The main graphic (panel B) shows what odorants are present in different extracts. What is the connection to the response properties of the palp neurons? Rather than odor present, shouldn't neuron response be mapped?

We have rewritten the part of the figure legend (“Presence/Absence matrix of the physiologically active compounds identified from the different headspace extracts for each MP-OSN in the GC-SSR experiments (i.e. each filled box represents not only the presence of this odor in a specific fruit, but also a physiological response in GC-SSR recordings).”) and hope that this decreased the confusion.

3) Figure 4. What odor concentrations do these represent?

We apologize and have added the information to the figure legend (“GC-SSR responses of the MP-OSNs best activators across the Ant-OSNs (n = 3, dilution, 10^-4^ in hexane).”)

4) Figure 5. Controls should have been compared to solvent odor controls, and not to zero. For example, maybe in their assay, the solvent itself leads to some attraction. Or if given a choice between two solvent background odors, the behavior would become highly variable (larger spread) such that the significantly different from zero data points would no longer be significant.

We agree that all responses should be compared to solvent odor controls. That is actually what we did (for each trap assay sample we count the number of flies in the trap filled with odor and solvent, and in the trap with solvent only. The attraction index (AI) is calculated as (number of flies in odor trap-number of flies in solvent trap)/total number of flies. An AI of 0 depicts equal attraction of odor and solvent, while an AI of 1 is a result of strong attraction to the odor (-1: strong repellency of the odor). Hence, by testing the result against 0, one tests, whether the odor is more (or less) attractive than the solvent control. To clarify we now tested against solvent control directly (resulting in the same level of significance) and have re-written the figure legend accordingly (“For T-maze and trap assays, the symbol * indicates significant differences from the solvent control (p < 0.05, Wilcoxon signed rank test, n = 10)”).

5) Figure 1. Unknown spelled Unknow in pb1B section.

Has been changed.

6) Genotypes, subsection “Complete genotypes of all strains used in this study”. w^1118^ instead of w^118^.

Has been changed.